# Design and Preparation of Double-Harmonic Piezoelectric Composite Lamination

**DOI:** 10.3390/ma15227959

**Published:** 2022-11-10

**Authors:** Chao Sun, Chao Zhong, Likun Wang, Lei Qin

**Affiliations:** Beijing Key Laboratory for Sensor, Beijing Information Science & Technology University, Beijing 100192, China

**Keywords:** double-harmonic, piezoelectric composite, lamination, bending vibration, finite element, conductance

## Abstract

In this work, a new type of double-harmonic piezoelectric composite laminated structure is designed. Two bending vibration frequencies are generated by designing the structure with non-equal length and non-equal width, and the response is relatively consistent at the frequency point of the double-harmonic vibration. Firstly, the finite element software ANSYS is used to establish the simulation model of double-harmonic piezoelectric composite lamination. Two bending vibration frequencies are generated by using the non-equal length structure design, and the variation law of the conductance curve with the laminated structure is analyzed. Then, according to this law, the structure is optimized, and a non-equal width structure is further proposed in this work. Different double-harmonic piezoelectric composite laminations are prepared for comparison. The simulation and experimental results show that the value of the corresponding conductance curve at the two vibration frequency points can be increased or reduced by changing the lamination width. Then, the same conductance peak can be obtained to have a relatively consistent response at the double-harmonic frequency point. This will provide a good choice for expanding the transducer bandwidth and developing the broadband energy collector.

## 1. Introduction

With the continuous development of science and technology, bending vibration lamination has become increasingly popular in water acoustic, air acoustic, energy harvesting, and so on. As a typical transducer, a laminated bending vibration transducer is usually composed of two piezoelectric material sheets. Its vibration principle is one of the pieces of the piezoelectric sheet at the same direction of the electric field and polarization direction; the other piezoelectric sheet has an electric field opposite to the direction of the polarization. When a driving voltage is applied, one piece expands, and the other contracts, causing the bending vibration of the double lamination [1]. At present, the circuit connection of the bending vibration lamination is mainly in series and parallel, of which the production process is more straightforward in series and slightly more complicated in parallel. The laminated bending vibration transducer has a simple structure, small size, and easy matching between air and water [1], which has been widely used in underwater acoustic detection, acoustic logging, industrial ultrasound, and other fields. In recent years, there have been many reports on it. Aronov studied a nonuniform piezoelectric disk laminated bending transducer’s resonant frequency and effective electromechanical coupling coefficient [2]. Zhang analyzed the effects of different boundary conditions and structural parameters on the resonant frequency and effective electromechanical coupling coefficient of a triple-laminated bending vibration transducer using the Rayleigh method theory [3]. Zheng et al. studied a triple-lamination-type dipole acoustic logging transducer that can measure the transverse wave velocity in fast and slow formations [4]. Xu et al. designed a triple-laminated bending-type hydrophone to achieve the high-sensitivity reception of low-frequency acoustic waves [5]. Wang produced a high-sensitivity double-laminated fluid acoustic emission transducer, which reduces the error in actual tool detection and provides some technical support for the automatic cutting process in the future [6]. Yuan proposed a low-frequency bender transmit transducer, which combines four different-sized tri-laminated elements to generate different resonant frequencies. The broadband transmissions of the transducer are realized by coupling multiple bending vibration modes [7].

The bending vibration lamination reported above is composed of piezoelectric ceramics. The excellent performance and scalability of piezoelectric composites have recently led many researchers to research piezoelectric composite lamination. Liu et al. proposed a piezoelectric composite bending vibration double lamination. By comparing the 2-2-type piezoelectric composite lamination, the 1-3-type piezoelectric composite lamination, and pure piezoelectric lamination, it was concluded that the 2-2 piezoelectric composites lamination has the most significant vibration amplitude and the lowest resonant frequency, which provides the basis for the laminated bending vibration transducer [8]. Lv et al. explored the bending vibration double-lamination of piezoelectric composites by changing the polymer type and boundary conditions. It was found that the performance of the silicone rubber-filled lamination was much more excellent than that of the epoxy resin-filled lamination. The vibration amplitude of the lamination was more extensive, and the resonance frequency was lower under fixed boundary conditions [9]. Lv et al. also proposed a piezoelectric composite bending vibration triple lamination and investigated the lamination’s performance under fixed and free boundary conditions. The conclusion showed that the performance of the lamination under fixed boundary conditions was better [10]. Chen et al. proposed a rectangular bipartite wafer based on a 2-2 piezoelectric composite, which has a larger vibration amplitude, lower resonant frequency, and better acoustic radiation capability compared with piezoelectric ceramics [11].

Compared with piezoelectric ceramics, the piezoelectric composite lamination has a larger vibration amplitude and a lower resonant frequency. However, so far, most of the studies on laminated bending vibration transducers are limited to single resonant frequency operations, and these laminations can only collect frequency points near their resonant frequencies. Therefore, a new double-harmonic piezoelectric composite-laminated structure with non-equal length and non-equal width is proposed in this work. The ceramic strip element in the laminated structure can generate two similar bending vibration frequencies. In particular, through the improvement of non-equal width, it obtains a more consistent response at the resonant frequency point. This laminated structure provides a useful exploration for the expansion of the working frequency band of the transducers and energy harvesters.

## 2. The Structure of Double-Harmonic Piezoelectric Composite Lamination

The structure of the double-harmonic piezoelectric composite lamination is shown in Figure 1. It consists of two layers of 2-2 piezoelectric composites with opposite polarization directions. In Figure 1, red and blue are piezoelectric ceramics, green is polymer, and gray is the electrode layer.

Figure 2 shows the geometrical dimensions of the double-harmonic piezoelectric composite lamination. In Figure 2, *l*_1_ and *w*_1_ represent the length and width of the long ceramic strip; *l*_2_ and *w*_2_ represent the length and width of the short ceramic strip; *w*_3_, *t*_1_, and *t* represent the width of the silicone rubber, the thickness of the ceramic strip, and the total thickness of the lamination.

## 3. Finite Element Simulation Analysis

To improve the computational efficiency, according to the periodicity of the double-harmonic piezoelectric composite lamination, this work selects one periodic unit of the lamination and uses the finite element analysis software ANSYS for simulation analysis. The finite element analysis is to divide the whole structure of the lamination into finite elements. The elements in each element represent the unknown function to be solved in the solution domain by the assumed approximate function to establish a multivariate system of equations, which is solved and summed using the computer [12].

Firstly, the material parameters of each phase material are defined. PZT-5A piezoelectric ceramic and 704 silicone rubber are used as a piezoelectric phase material and a polymer phase material. Solid 5 is selected as the unit type of the piezoelectric phase and solid 45 as the unit type of the polymer phase. Detailed material parameters are shown in Table 1 [13,14,15,16]. The “SWEEP” function is used to divide the finite element mesh, and the side length of each element is 1 mm. A voltage of 1 V and 0 V is added to the upper and lower surfaces of the divided finite element model. The harmonic response analysis is selected under the free boundary condition: the frequency range is set as 3000–7000 Hz, the number of steps is 200, the step length of each step is 20 Hz, and the damping coefficient is set as 0.02. The conductance curve and vibration mode of lamination can be obtained by harmonic response analysis. As shown in Figure 3, there are two resonance peaks in the conductance curve. The vibration mode is shown in Figure 4. It can be seen that the vibration mode is transverse bending vibration.

### 3.1. Simulation Analysis of Double-Harmonic Piezoelectric Composite Lamination

To study the influence of the non-equal length structure of ceramic strips on the bending vibration frequency of lamination, the finite element simulation method is used to study the resonant frequency and conductance of the lamination. Keep the length of one ceramic strip unchanged in the lamination and change the length of the other ceramic strip. In the work, the thickness of the ceramic strip *t*_1_ is 2 mm (the total thickness of the lamination *t* is 4 mm), the widths *w*_1_ and *w*_2_ are both 2 mm, and the width of the silicone rubber *w*_3_ is 1 mm. The length *l*_1_ of the long ceramic strip is always 50 mm, and the length *l*_2_ of the short ceramic strip changes from 45–50 mm in turn. The obtained conductance curves are shown in Figure 5. The length *l*_2_ of the short ceramic strip is always 50 mm, and the length *l*_1_ of the long ceramic strip changes from 50–55 mm. The obtained conductance curves are shown in Figure 6.

Figure 5a shows the conductance curves of *l*_1_ unchanged and *l*_2_ gradually decreasing from 50 mm to 49 mm. According to Figure 5a, there are obvious double peaks in the conductance curve after *l*_2_ becomes less than 49.6 mm. The peak value and frequency of the second resonant peak on the conductance curve show a slow upward trend when *l*_2_ is further reduced.

Figure 5b shows the conductance curves of *l*_1_ unchanged and *l*_2_ gradually decreasing from 50 mm to 49 mm. According to Figure 5b, it is shown that with the decrease in *l*_2_, the first peak shows a downward trend, and the corresponding frequency gradually increases. In contrast, the second peak shows an upward trend, and the corresponding frequency rises significantly.

Figure 6a shows the conductance curves of *l*_2_ unchanged and *l*_1_ gradually increasing from 50 mm to 51 mm. From Figure 6a, there are obvious double peaks in the conductance curve after *l*_1_ is greater than 50.4 mm. When *l*_1_ further increases, the peak of the second resonant peak on the conductance curve shows a slow upward trend and the corresponding frequency shows a slow downward trend.

Figure 6b shows the conductance curves of *l*_2_ unchanged and *l*_1_ gradually increasing from 51 mm to 55 mm. From Figure 6b, it is shown that with the increase in *l*_1_, the first peak shows a downward trend, and the corresponding frequency drops sharply, while the second peak shows an upward trend, and the corresponding frequency decreases gradually.

The above analysis shows that when the difference between the lengths of two ceramic strips (*l*_1_–*l*_2_) of the lamination is 0.4 mm, the lamination will produce two different vibration frequencies. When the value of *l*_1_–*l*_2_ is greater than 0.4 mm, the conductance curve begins to appear with two obvious resonance peaks that produce the double-harmonic bending vibration. As the value of *l*_1_–*l*_2_ gradually increases, the first peak value of the conductance curve gradually decreases, and the second peak value gradually increases. However, the peaks of the two resonance peaks are still quite different, and the lamination cannot obtain a consistent response. Therefore, the laminated structure needs to be further improved.

### 3.2. Simulation Analysis of Improved Double-Harmonic Piezoelectric Composite Lamination

To solve the above problems, this work proposes a non-equal width structure design to improve the scheme by increasing *w*_2_ to increase the second resonance peak or reducing *w*_1_ to decrease the first resonance peak. By changing the width of the lamination, the two peaks of the conductance curve are basically equal.

The lamination of ceramic strips with two lengths *l*_1_ = 55 mm and *l*_2_ = 50 mm is selected for simulation verification. With *w*_2_ unchanged, *w*_1_ is gradually reduced from 2 mm to 1 mm. The change curve is shown in Figure 7a. Figure 7a shows that as *w*_1_ decreases, the first peak decreases significantly, and the corresponding frequency increases. When *w*_1_ reduces to half of *w*_2_, the first peak is 1.5 times that of the second peak and cannot reach an agreement. Since the width of the ceramic strip selected in this work is narrow, it will be difficult to continue to shorten the width of the ceramic strip in the subsequent process. Therefore, on the basis of reducing *w*_1_, the method of increasing *w*_2_ is adopted for improvement. The *w*_1_ is selected as 1 mm, the *w*_2_ is gradually increased from 2 mm to 2.5 mm, and the change curve is shown in Figure 7b. It can be seen from Figure 7b that the second peak value rises sharply with the increase in *w*_2_. When *w*_2_ increases to 2.5 mm, the two conductance peaks are basically the same.

## 4. Experimental Preparation and Performance Testing of Double-Harmonic Piezoelectric Composite Lamination

Based on the finite element simulation, a single period of the double-harmonic piezoelectric composite lamination is fabricated in this work. The preparation process is shown in Figure 8. The specific preparation steps are as follows:(1)A block of 5 mm-thickness piezoelectric ceramic PZT-5A is selected and carved into a non-equal shape using a CNC engraving machine.(2)The carved ceramic is cut, but the substrate is retained.(3)The silicone rubber is infused into the slit of the cut piezoelectric ceramic.(4)The previously retained substrate is removed, but the positive electrode is retained.(5)The two polished positive surfaces of the ceramic are bonded with epoxy resin.(6)The magnetron sputterer is used to sputter the two negative surfaces without electrodes on the top and bottom.

As shown in Figure 9, the piezoelectric composite laminations created by the above preparation method are Sample A, Sample B, Sample C, and Sample D. The thickness of all samples *t* (including the thickness of silicone rubber in the sample) is 4 mm. Sample A is a double-harmonic piezoelectric composite lamination with dimensions of *l*_1_ = 50 mm, *l*_2_ = 47 mm, *w*_1_ = 2 mm, and *w*_2_ = 2 mm. Sample B is a conventional piezoelectric composite lamination with dimensions of *l*_1_ = 50 mm, *l*_2_ = 50 mm, *w*_1_ = 2 mm, and *w*_2_ = 2 mm. Sample C is a double-harmonic piezoelectric composite lamination with dimensions of *l*_1_ = 55 mm, *l*_2_ = 50 mm, *w*_1_ = 2 mm, and *w*_2_ = 2 mm. Sample D is a modified double-harmonic piezoelectric composite lamination with dimensions of *l*_1_ = 55 mm, *l*_2_ = 50 mm, *w*_1_ = 1 mm, and *w*_2_ = 2.5 mm.

The prepared double-harmonic piezoelectric composite laminated samples are tested with an impedance analyzer (Agilent 4294A, Agilent Technologies, Inc., Santa Clara, CA, USA). The test frequency range of the impedance analyzer is 40–110 MHz, and the resolution is 1 mHz. The samples are connected to the impedance analyzer according to the test method shown in Figure 10. The high voltage end and low voltage end of the impedance analyzer are, respectively, connected to the upper and lower surfaces of the lamination. The measured conductance curve is shown in Figure 11.

Figure 11 shows the test conductance curve of the double-harmonic laminated sample. Figure 11a shows the comparative conductance curves of Sample A and Sample B, and Figure 11b shows the comparative conductance curves of Sample C and Sample B. It can be seen from Figure 11a,b that the conductance curves of samples A and C both show double peaks, which means that the laminations can appear as double-harmonic vibrations by changing the different lengths of the porcelain strips. Figure 11c shows the comparative conductance curves of Sample D and Sample C. Figure 11c shows that the two peaks of Sample D are similar, and the improved double-harmonic piezoelectric composite lamination can obtain relatively consistent responses.

Comparing Figure 6, Figure 7 and Figure 8 and Figure 11, it can be found that the finite element analysis results are consistent with the experimental tests, but there are also certain deviations. There may be two reasons for the error: (1) The environment simulated by ANSYS is too idealized, while the actual experimental environment does not reach the absolute ideal conditions. (2) There are errors in the material parameters for the experiments and the simulations, which need to be further investigated.

## 5. Discussion

This work introduces a double-harmonic piezoelectric composite lamination. The results of the simulation and experiment show that: (1) Compared with the lamination that generates a single bending frequency, the double-harmonic composite lamination proposed in this work has been improved in structure. The lamination can generate two similar bending vibration frequencies and obtain a consistent response. It can be used to collect multiple operating frequency points in the energy harvester. (2) Compared with the transducer composed of several different sizes of triple laminations, the lamination proposed in this work is small in size, easy to fabricate, and can prevent experimental errors in the splicing process. It can be applied to broadband transmitting transducers to improve the bandwidth.

## 6. Conclusions

In this work, a new type of double-harmonic piezoelectric composite lamination is proposed by using the finite element simulation method and the design of non-equal length and non-equal width structure. After comparing the simulation results of double-harmonic piezoelectric composite laminations many times, the optimal length and width of the laminated ceramic strips are determined. The specific results are as follows: (1) In the non-equal length design, the length of one of the ceramic strips is fixed at 50 mm. When the difference between the other ceramic strip and its length is greater than 0.4 mm, the lamination can generate double-harmonic bending vibration. (2) Based on the non-equal length design, the peak value corresponding to the double-harmonic frequencies can be made close by changing the width of the ceramic strip, and a more consistent response can be obtained. To verify the finite element simulation results, a double-harmonic piezoelectric composite lamination sample is prepared and tested. The results are basically consistent with the simulation results. The experimental results show that the double-harmonic bending vibration can be generated by changing the length of the porcelain strip, and the lamination can receive a more consistent response by further changing the width of the porcelain strip. Therefore, the double-harmonic piezoelectric composite lamination proposed in this work expands from a single working frequency of traditional laminations to multiple working frequencies. It not only retains the miniaturization and low-frequency characteristics of traditional lamination but also can broaden the working frequency band of the laminated bending vibration transducer and increase the collection frequency of the energy harvester. It has high practical value in the field of underwater acoustic, air acoustic, and energy harvesting.

## Figures and Tables

**Figure 1 materials-15-07959-f001:**
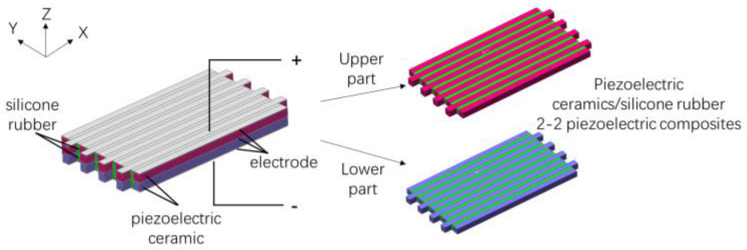
Structure of the double-harmonic piezoelectric composite lamination.

**Figure 2 materials-15-07959-f002:**
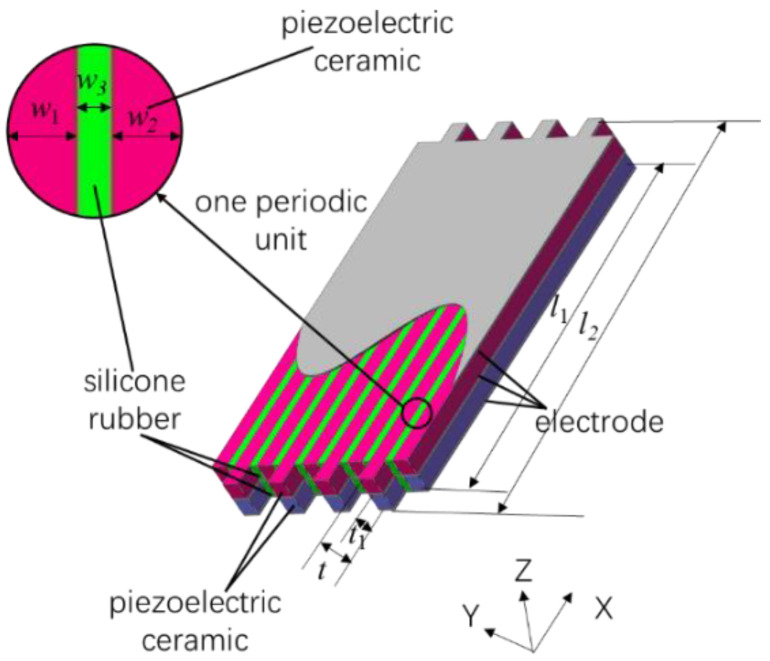
The geometry of double-harmonic piezoelectric composite lamination.

**Figure 3 materials-15-07959-f003:**
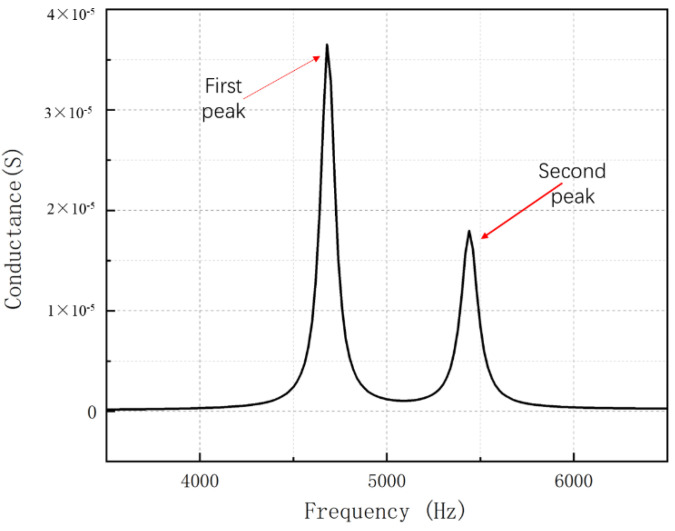
Conductance curve of the double−harmonic piezoelectric composite lamination (*l*_1_ = 50 mm, *l*_2_ = 46.5 mm).

**Figure 4 materials-15-07959-f004:**
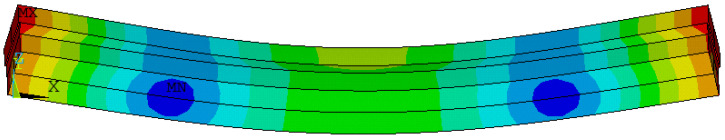
Vibration mode of the double−harmonic piezoelectric composite lamination (*l*_1_ = 50.3 mm, *l*_2_ = 50 mm).

**Figure 5 materials-15-07959-f005:**
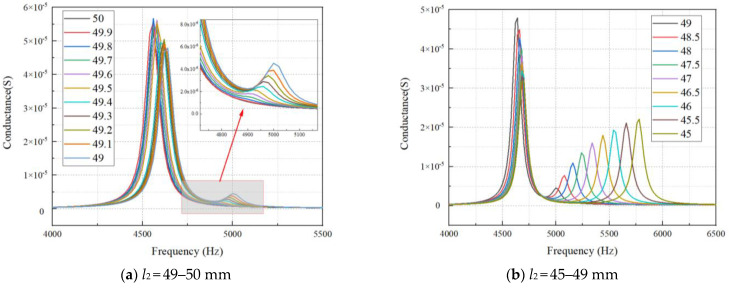
Conductance curve with different *l*_2_ (*l*_1_ = 50 mm).

**Figure 6 materials-15-07959-f006:**
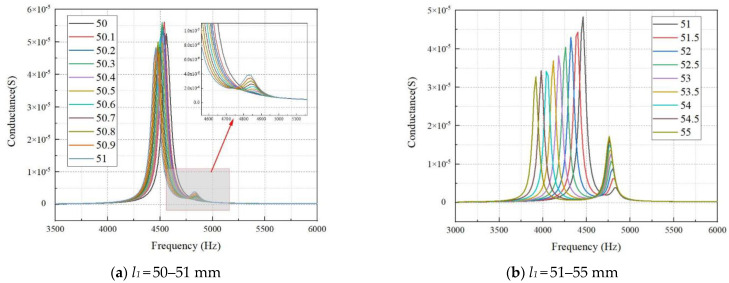
Conductance curve with different *l*_1_ (*l*_2_ = 50 mm).

**Figure 7 materials-15-07959-f007:**
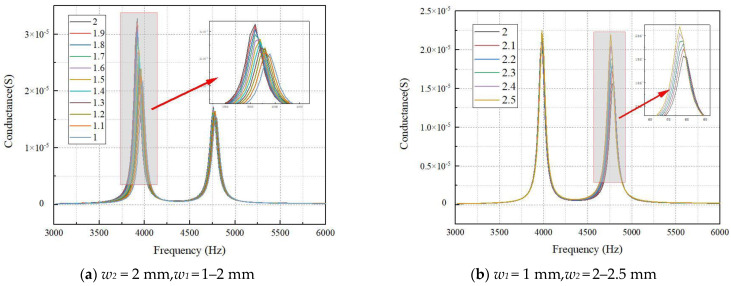
Conductance curves in different widths.

**Figure 8 materials-15-07959-f008:**
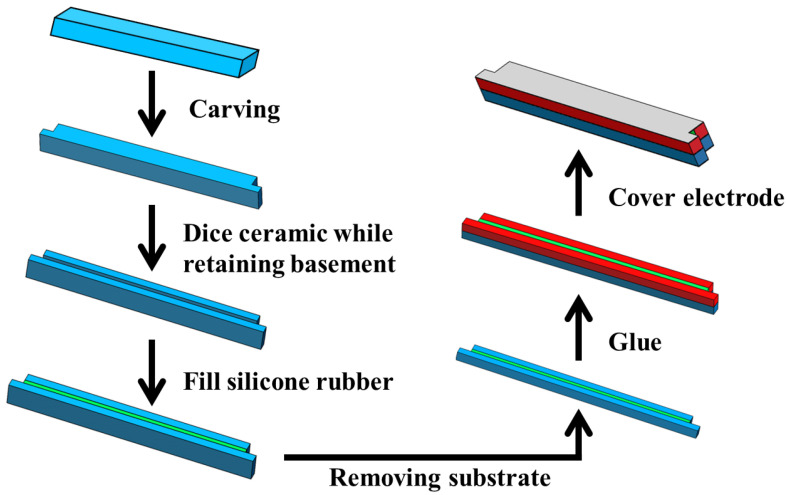
The preparation process of the double-harmonic piezoelectric composite lamination.

**Figure 9 materials-15-07959-f009:**
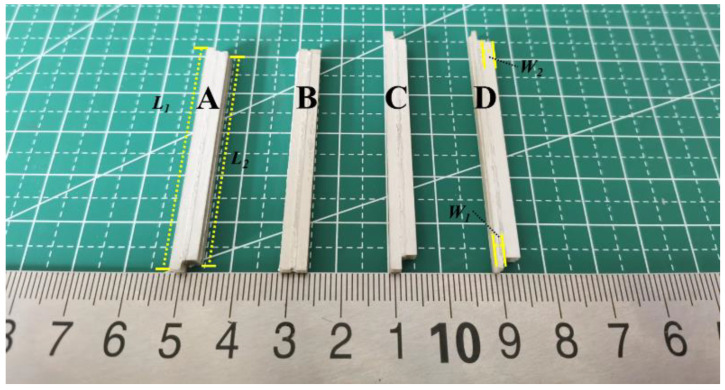
The sample of double-harmonic piezoelectric composite lamination.

**Figure 10 materials-15-07959-f010:**
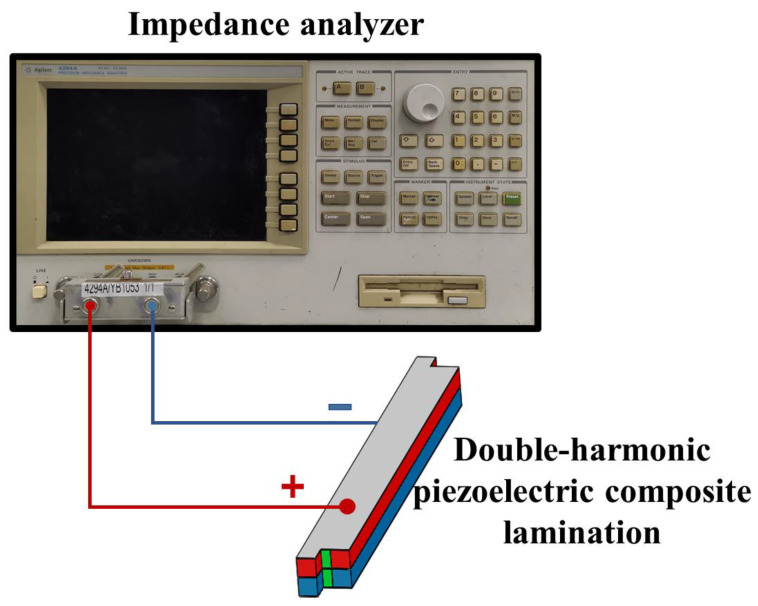
Diagram of the test method for laminated samples.

**Figure 11 materials-15-07959-f011:**
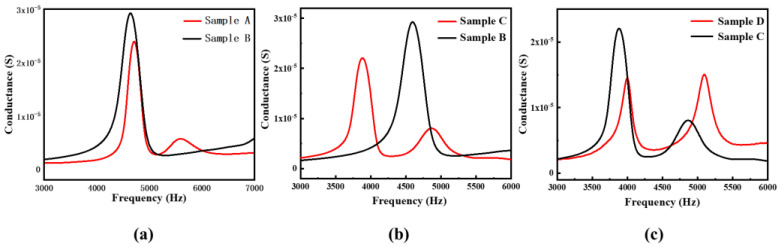
Test results of conductance curve. (**a**) The comparative conductance curves of Sample A and Sample B. (**b**) The comparative conductance curves of Sample B and Sample C. (**c**) The comparative conductance curves of Sample C and Sample D.

**Table 1 materials-15-07959-t001:** Parameters of the piezoelectric ceramics, PZT-5A, and silicone rubber.

Parameter	PZT-5A	Silicone Rubber
*ρ* (kg/m^3^)	7750	1050
*c*^E^_11_ (10^10^ N/m^2^)	12.1	0.004
*c*^E^_12_ (10^10^ N/m^2^)	7.54	0.0023
*c*^E^_13_ (10^10^ N/m^2^)	7.52	/
*c*^E^_33_ (10^10^ N/m^2^)	11.1	/
*s*^E^_11_ (10^−12^ m^2^/N)	16.4	4 × 10^5^
*s*^E^_12_ (10^−12^ m^2^/N)	−5.74	2.3 × 10^5^
*s*^E^_13_ (10^−12^ m^2^/N)	−7.22	/
*s*^E^_33_ (10^−12^ m^2^/N)	18.8	/
*d*_31_ (10^−12^ C/N)	−171	/
*d*_33_ (10^−12^ C/N)	470	/
*e*_31_ (C/m^2^)	−5.4	/
*e*_33_ (C/m^2^)	15.8	/
*ε*^S^_33_/ε_0_	830	3.3
*ε*^T^_33_/ε_0_	1700	3.3

## Data Availability

The data presented in this study are available on request from the corresponding author.

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
