# Peer review of "Design and Preparation of Double-Harmonic Piezoelectric Composite Lamination"

_materials, 2022, doi:10.3390/ma15227959_

Round 1
Reviewer 1 Report
In my opinion, the manuscript is suitable for publication in Materials journal but Authors must complete a major revision. Manuscript should be revised according to following comments:
1. Chapter “Introduction” must be improved:
a) the terms used are incorrect, e.g. instead of "vibration displacement" there should be "vibration amplitude". The authors must correct all such terms,
b) in lines 74-75 the authors write that "most of the studies on laminated bending vibration transducers have been limited to single resonant frequency operation". Please list examples from this minority that deal with laminate with more than one resonant frequency,
c) please explain what the authors bring to the stat of art in the area of laminate structures in terms of materials.
2. Chapter “Double-harmonic piezoelectric composite laminated structure” must be improved:
a) please present the differences between the proposed structure from Fig. 1 and the commercial Macro Fiber Composite sold by Smart Materials Corp.
3. Chapter “Finite element simulation analysis” must be improved:
a) at the beginning of this chapter, the methodology of numerical research should be presented,
b) it is necessary to explain what the figures 2 and 3 bring in. In my opinion, they do not bring any knowledge about the topic of the article.
4. Chapter “Experimental preparation and performance testing of double-harmonic lamination” must be improved:
a) the description of the research methodology should be supplemented with:
- a diagram of the laboratory stand,
- sensor parameters (resolution, measuring range, etc.).
5. Authors must add a chapter with a discussion of the results. In this chapter, the authors should relate their obtained research results to results of other researchers.
6. Chapter “Conclusions” must be improved:
a) in this chapter the authors summarize what they did. There is no clear indication of what, from the scientific point of view, is a new achievement in relation to the existing state of art.
Reviewer 2 Report
Comments:
Major Revisions for reconsideration.
1. “In conclusion“ on page 2 should not be presented in the introduction part, please modify it.
2. The structure of the manuscript is suggested to be corrected? The headline of 2, 3 should be “experimental“ and “results and discussions“
3. There are many typo errors and awkward expression in English.
4. The authors should provide the detailed information of the materials used in this study.
5. The l1, l2, w1, w2 should be labeled in Fig. 10.
6. the author claimed that all samples have a thickness of 4mm. For samples A, b and C, w1, and w2 are 2 mm, respectively. Then, how about the thickness of the silicon rubber?
Reviewer 3 Report
Dear authors, I consider that your manuscript needs major revision. Please see the remarks presented in the attached review document.

Round 2
Reviewer 1 Report
I accept in present form.
Reviewer 2 Report
the authors have revised the manuscript, it can be accepted now.
Reviewer 3 Report
The authors have significantly revised and improved the manuscript. I think the manuscript can be accepted for publication.